# HCN1 is a primary HCN Pacemaker Channel in Neurons

Uta Enke [1], Andrea Schweinitz[1], Debanjan Tewari[1], Christian Sattler [1], Ralf Schmauder [1], Christoph Schmidt-Hieber [2] & Klaus Benndorf [1] ✉

Rhythmic activity of specialized pacemaker neurons in the brain is necessary to control alertness and circadian timing. Four HCN channels have been identified to generate the pacemaker current $I_h$ or $I_q$, differing in activation speed, voltage dependence, single-channel conductance, and cAMP sensitivity. Here we show the time-resolved operation of single HCN1, HCN2 and HCN4 channels during the pacemaker depolarization using a dynamic neuronal action potential clamp at femtosiemens resolution. All channels produce a relevant open probability during pacemaker depolarization. However, only mHCN1 channels are significantly activated and deactivated in action potential cycles whereas the gating in mHCN2 and mHCN4 channels is at best barely resolvable and too slow. Simulations suggest that the role of HCN1 channels is to trigger the initial neuronal pacemaker depolarization before other depolarizing conductances take over this role. In conclusion, mHCN1 channels are the primary HCN pacemaker channels that operate as trigger channels for pacemaking.

Hyperpolarization-activated cyclic nucleotide-modulated (HCN) ion channels are thought to generate the pacemaker current $I_h$ or $I_q$ in specialized neurons of the brain, thereby producing rhythmic electrical activity[1–7]. They have also been implicated in other functions of the central nervous system, such as modulating working memory and synaptic transmission[2]. In mammalians, there are four subunit isoforms HCN1-4[1,8,9], all of which can form functional homotetrameric channels[10–12]. Upon sympathetic stimulation, the second messenger cAMP binds to the cyclic nucleotide binding domains in each subunit, thereby increasing the pacemaking frequency. The four homomeric channels differ functionally in their activation speed[12,13] and the intensity of the cAMP effect[1,12–18]. The single-channel conductance is exceptionally small near 1 pS for all isoforms but also differs up to threefold[13].

The voltage of half maximum activation was found to be least negative for HCN1[19,20], intermediate for HCN2[18,21–23] and most negative for HCN4[13,21,24,25], but shows substantial variation for each isoform across different studies. This variability seems to depend critically on the expression system, the recording technique and the used pulse protocols. One reason for this variability can be the mode-shift phenomenon, describing a dependence of the working point for the action of HCN channels on the history of the applied pulses[26–28]. Beyond this variability, activation of all HCN channels occurs at voltages much more negative than the −55 to −80 mV range of neuronal pacemaker potentials[29–31], and the activation kinetics in voltage clamp experiments are much slower than typical action potential frequencies between 10 to 50 Hz. These findings call into question their role in neuronal pacemaking.

To address this issue, it is necessary to dynamically follow the activity of HCN channels during sequences of action potentials. A powerful method to achieve this is the action-potential clamp in which the voltage trajectory of an action potential is played back to the cell as the voltage command[32–34]. In macroscopic recordings, the activity of the desired channels can be extracted using selective and efficient blockers[31,35–37], which is, however, an idealized assumption. Here we show at single-channel resolution the time-dependent open probability for mHCN1, mHCN2 and mHCN4 channels during pacemaker depolarization of typical neuronal action potentials by using a dynamic

[1]Institut für Physiologie II, Universitätsklinikum Jena, Friedrich-Schiller-Universität Jena, Jena, Germany. [2]Institut für Physiologie I, Universitätsklinikum Jena, Friedrich-Schiller-Universität Jena, Jena, Germany. ✉e-mail: Klaus.Benndorf@med.uni-jena.de

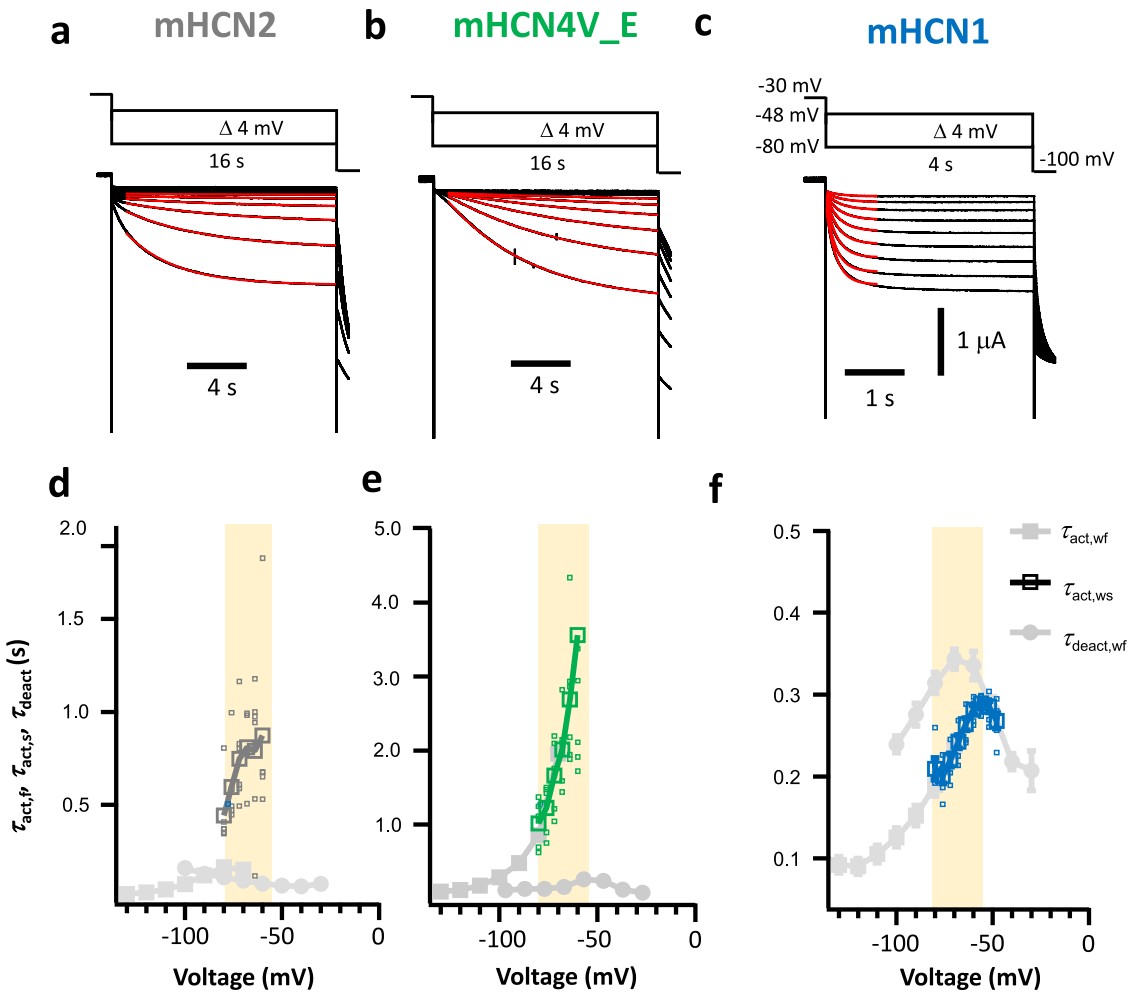

**Fig. 1 | Activation kinetics at pacemaker voltages with slow protocol. a–c** To analyze the activation speed in the pacemaker voltage range, hyperpolarizing pulses of long duration were applied. Pulsing frequency: mHCN1, 0.0833 Hz; mHCN2 and mHCN4V_E, 0.03125 Hz. **d–f** Activation time constant $\tau_{act,ws}$ as function of voltage determined by an exponential fit. The individual data points and the respective means (mHCN1: $n = 6$; mHCN2: $n = 6$; mHCN4V_E: $n = 6$) are shown. For comparison, the means±SEM of $\tau_{act,wf}$ and $\tau_{deact,wf}$ (Supplementary Fig. 1) were included in gray color. The yellow rectangle illustrates the pacemaker voltage range between −80 and −55 mV.

action-potential clamp. Only mHCN1 is sufficiently rapid to be a genuine pacemaker channel at typical neuronal firing rates, whereas mHCN2 and mHCN4 are not.

## Results

### Activation and deactivation in whole oocytes

With the two-electrode voltage clamp (TEVC) and a standard voltage protocol (Methods), macroscopic currents were recorded from mHCN1, mHCN2 and mHCN4V_E (mHCN4V487E) channels and fitted with Eq. (3). The activation time constants $\tau_{act,wf}$ showed the characteristic exponential rise to less negative voltages (Supplementary Fig. 1a–c, g)[12,13,15,18]. We used the mutant mHCN4V_E instead of mHCN4 because its single-channel conductance approximates the larger single-channel conductance of mHCN2[13] while the activation gating resembled mHCN4 (Supplementary Fig. 2a–d). The use of this mutant uniquely allowed us to identify the unitary currents of mHCN4 channel gating at pacemaker voltages.

Deactivation kinetics following a pulse to −130 mV were fitted by an exponential function yielding the time constant $\tau_{deact,wf}$ (Supplementary Fig. 1d–f). Plotting $\tau_{deact,wf}$ together with $\tau_{act,wf}$ (Supplementary Fig. 1g) suggests larger values at intermediate voltages for all three channels[38,39]. Because with the standard voltage protocol, activation in the pacemaker range was incomplete[40], we applied in this voltage range also a protocol with longer pulses (Methods) (Fig. 1a–c). The

expectedly small currents were fitted by Eq. (3) between −80 and −60 mV for the three channels, yielding the time constant $\tau_{act,ws}$.

This leads to two main results. First, for all channels, the differences between the time constants at overlapping voltages indicate a more complex gating process than describable by one closed and one open state, questioning the sense of applying monoexponential fits to interpret the channel gating beyond a phenomenological description. Second, in the pacemaker voltage range, $\tau_{act,ws}$ is slow for all channels (Fig. 1d–f; 0.5-1 s for HCN2, 1-4 s for mHCN4V_E, 0.2-0.3 s for mHCN1) which conflicts with the idea that the channels generate neuronal AP frequencies of 10 or 50 Hz by activation and deactivation.

However, it is a priori not clear that the time constants $\tau_{act,ws}$, $\tau_{act,wf}$ and $\tau_{deact,wf}$ determined by hyperpolarizing rectangular pulses (Fig. 1d–f; and Supplementary Fig. 1g) mirror indeed key features of HCN channels working in dynamic equilibria of action potential trains because repetitive activation and deactivation could drive the channels into other functional states. Consequently, we measured the open probability $P_o(t)$ directly in trains of pacemaker action potentials using a dynamic action potential clamp at single-channel resolution.

### Fast single-channel action potential clamp at 10 Hz

Currents were studied in cell-attached patches of *Xenopus* oocytes using as command voltage a reconstructed action potential time course from a spontaneously firing suprachiasmatic neuron at room

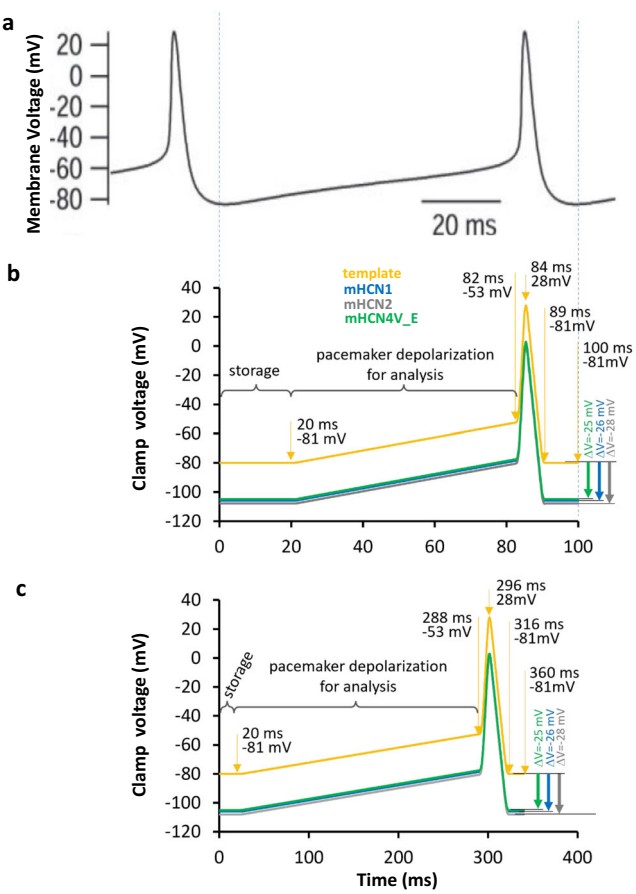

**Fig. 2 | Voltage protocols used for the AP potential clamp. a** Spontaneous action potential of a suprachiasmatic neuron at room temperature obtained from Jackson and coworkers[31] (with permission, Copyright 2004 Society for Neuroscience). The time course was used as template to design idealized action potentials by a sequence of voltage ramps. **b** Fast AP frequency of 10 Hz. Action potentials used as command for the dynamic action potential clamp. The ochre trace shows the idealized action potential as obtained from the trace in a used as command voltage. Voltages and times are indicated. Because the voltage of activation was shifted ($\Delta V_x = V_{hpx} - V_{hwx}$) by −26 mV (mHCN1), −28 mV (mHCN2) and −25 mV (mHCN4V_E) to more negative values (Supplementary Table 1), the actual command traces were obtained by adding these voltages to the ochre idealized action potential time course. **c** Slow AP frequency of 2.77 Hz used as command potential. The ochre trace shows the action potential time course as obtained from slowing the fast time course (see b) three times while leaving the time interval required for storage at 20 ms. The resulting pulsing rate is 2.77 Hz. Voltages and times are indicated. The shift of voltage activation was the same as in b.

temperature[31] (see Methods; Fig. 2a), resulting in our 'fast AP frequency protocol (f)' (Fig. 2b). With respect to the TEVC data (w), steady-state activation in the patches was shifted by unknown reasons to more negative voltages for all channels (x = 1,2 or 4)[13]. We therefore determined steady-state activation also in patches with ensemble currents (p) (Supplementary Fig. 3; Supplementary Table 1) and corrected the command voltages by $\Delta V_x = V_{hwx} - V_{hpx}$, yielding the action potential time courses $AP(t)_{xf}$ (Fig. 2b; Methods). $AP(t)_{xf}$ was used as command voltage generating $I(t)_{xf}$. At least 50 $AP(t)_{xf}$ were applied repetitively in trains (Supplementary Fig. 4a). Each AP train was preceded by a pulse of 4 s to −30 mV to invoke deactivation.

For mHCN2, Supplementary Fig. 5a–c illustrates raw traces, additionally filtered traces and null-subtracted traces for $I(t)_{2f}$. Discrete levels of the unitary currents become directly visible after appropriate filtering (Fig. 3a). Because in $I(t)_{2f}$ the voltage changes along the

clamped $AP(t)_{2f}$, the conductance $G(t)_{2f}$ in the patch was calculated by Eq. (4) for the pacemaker depolarization interval. The unitary conductance is near the expected value of 1.5 pS (Fig. 3b)[13]. From $G(t)_{2f}$ the open probability along the pacemaker depolarization, $P_o(t)_{2f}$, was obtained by Eq. (5). The result was that both the 15 individual $P_o(t)_{2f}$ time courses and their average, $\langle P_o(t)_{2f} \rangle$ were constant over the interval of pacemaker depolarization in the considered time window (Fig. 3c). Longer AP trains produced the largest $P_o(t)_{2f}$ values (Supplementary Table 2). The constancy of $P_o(t)_{2f}$ is in contrast to the expectation for a pacemaker channel that $P_o(t)$ first increases and then decreases during the pacemaker depolarization, ruling out mHCN2 channels as genuine pacemaker channel at the AP frequency of 10 Hz.

To consider slower processes of the HCN channel activity, the average conductance of the pacemaker interval, $\langle G_{xfp} \rangle(t)$, was calculated and plotted along the AP train. For the fast AP train in mHCN2, $\langle G_{2fp} \rangle(t)$ produced a markedly slow recovery from deactivation over many seconds, overlaid by a stochastic switching of the channels (Fig. 3d). This slowness of the recovery from deactivation fits to the observation that longer AP trains generate larger $P_o(t)_{2f}$ values (Fig. 3c). We see the slow recovery from deactivation as a slow relaxation process of the channel protein, which is presumably the cause for the constancy of $P_o(t)_{2f}$, i.e., the absence of a gating, during the pacemaker depolarization.

The experiments were then repeated for the slower mHCN4V_E channels (Supplementary Fig. 5d–f; and Fig. 3e–h) using a similar voltage shift $\Delta V_4$ as for mHCN2 (Supplementary Table 1). The current traces resemble those in mHCN2 channels. In the $I(t)_{4f}$ and $G(t)_{4f}$ plots, discrete levels are again visible, matching in amplitudes our previous results[13]. In the considered time window of pacemaker depolarization, both the individual time courses $P_o(t)_{4f}$ and their average $\langle P_o(t)_{4f} \rangle$ are constant again (Fig. 3g), confirming that also HCN4V_E channels do not gate during the pacemaker depolarization. Also, longer AP trains generate larger $P_o(t)_{4f}$ values (Fig. 3g; and Supplementary Table 3) and $\langle G_{4fp} \rangle(t)$ produces a markedly slow recovery from deactivation over many seconds (Fig. 3h). As for mHCN2 channels, the slow recovery from deactivation in mHCN4V_E channels seems to reflect a slow relaxation process.

The experiments were repeated then with rapidly gating mHCN1 channels. Using a voltage shift $\Delta V_1$, similar to the other two channels (Supplementary Table 1), the fast action potentials $AP(t)_{1f}$ were specified (Fig. 2b). The experiments were a priori more challenging because the conductance of mHCN1 channels of 0.84 pS is only about half of that in mHCN2[13]. However, our resolution sufficed to run the same analysis as for the other two isoforms (Supplementary Fig. 5g–i; and Fig. 3i–l).

The $G(t)_{1f}$ plot (Fig. 3j) confirms the previously determined conductance. In the shown patch, only either 0, 1, or 2 channels out of 3 channels were active at pacemaker depolarizations. Also, for the fastest HCN isoform, neither the individual $P_o(t)_{1f}$ time courses nor their average $\langle P_o(t)_{1f} \rangle$ reveal a time dependence in the considered time window of pacemaker depolarization (Fig. 3k; for statistics see Supplementary Table 4).

In strong contrast to mHCN2 and mHCN4V_E, $\langle G_{1fp} \rangle(t)$ provided solely a stochastic fluctuation along the 50 successive APs (Fig. 3l). Hence, the molecular processes underlying the slow recovery in mHCN2 and mHCN4V_E channels was not observed in individual experiments with mHCN1 channels.

## Slow single-channel action potential clamp at 2.77 Hz

To test if a slower AP frequency leads to an identifiable activation and deactivation gating during the pacemaker depolarization, we applied a 'slow AP frequency protocol (s)' with AP trains at 2.77 Hz (Supplementary Fig. 4b and Fig. 2c), a frequency that is also well in the range of autorhythmic firing at room temperature, as e.g., in rat cerebellar Golgi cells[41]. The analyzed pacemaker depolarization lasted 268 ms (Methods). The experiments were repeated accordingly with $AP(t)_{xs}$, providing $I(t)_{xs}$ (Supplementary Fig. 6c, f, i), $G(t)_{xs}$ (Fig. 4b, f, j), $\langle P_o(t)_{xs} \rangle$

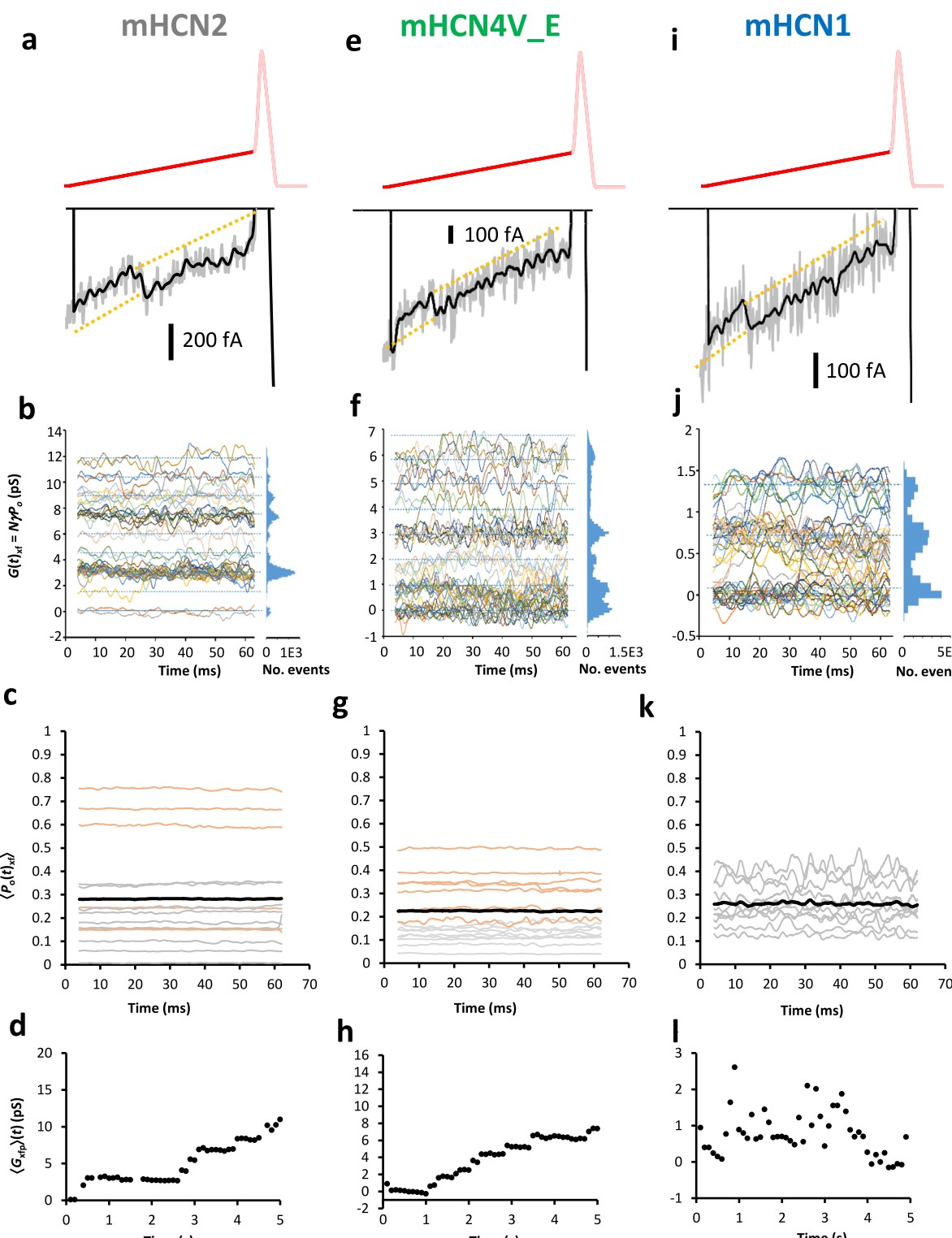

# AP frequency 10 Hz

(Fig. 4c, g, k), and $\langle G_{xsp}\rangle(t)$ (Fig. 4d, h, l). As for the fast AP frequency, discrete levels could be identified easily already in the raw current recordings, after appropriate filtering and the null-subtracted traces, $I(t)_{xs}$, as well as in the representative recordings of Fig. 4a, e, i. The $G(t)_{xs}$ plots confirm the discrete levels (Fig. 4b, f, j).

For mHCN2, the individual $P_o(t)_{2s}$ time courses suggest also time independence (Fig. 4c). The respective average $\langle P_o(t)_{2s}\rangle$ has a value

near 0.21. However, the lower noise in the average time course does reveal a small activation in the first part of the 268-ms time window of pacemaker depolarization (inset in Fig. 4c). This activation time course, $\Delta P_o(t)$, was quantified by fitting the exponential equation

$$\Delta P_o(t) = P_{o,0} + \Delta P_{o,p}[1 - \exp(-t/\tau_{act,p})]. \tag{1}$$

**Fig. 3 | Activation of mHCN channels at the fast AP frequency of 10 Hz.** Three representative experiments are illustrated. The current recordings followed a 4 s pulse to −30 mV. **a–d** mHCN2. a, Current trace with one channel opening. **b** Conductance $G(t)_{2f}$ as function of time. The dashed horizontal lines and the amplitude histograms right suggest equidistance of the levels near 1.5 pS. 45 traces are overlaid. **c** 15 $P_o(t)_{2f}$ measurements (gray: AP trains with 50 APs; light brown: longer AP trains; black: $\langle P_o(t)_{2f}\rangle$ over all 15 measurements). For statistics see Supplementary Table 2. $P_o(t)_{2f}$ is constant over the time interval. **d** Average conductance $\langle G_{2fp}\rangle(t)$ during pacemaker depolarization. Representative experiment. $\langle G_{2fp}\rangle(t)$ rises slowly over seconds while the stochastic activity of the single channels superimposes. **e–h** mHCN4V_E, analogue to (**a–d**). **e** Current trace with one channel opening. **f** Conductance $G(t)_{4f}$ as function of time. 50 traces are overlaid. g, 16 $P_o(t)_{4f}$ measurements; analogue to c with Supplementary Table 3. **h** $\langle G_{4fp}\rangle(t)$ during pacemaker depolarization. $\langle G_{4fp}\rangle(t)$ also rises slowly over seconds. **i–l** mHCN1, analogue to (**a–d**). **i** Current trace with one channel opening. **j** Conductance $G(t)_{1f}$ as function of time. 50 traces are overlaid. **k** 10 $P_o(t)_{1f}$ measurements; analogue to c with Supplementary Table 4. l, $\langle G_{1fp}\rangle(t)$ during pacemaker depolarization. Representative experiment. $\langle G_{1fp}\rangle(t)$ reveals a stochastic activity of the time interval and deviates fundamentally from the slowly rising $\langle G_{2fp}\rangle(t)$ and $\langle G_{4fp}\rangle(t)$.

$P_{o,0}$ is the $P_o$ value at the start of the time interval, $\Delta P_{o,p}$ the amplitude of the time-dependent component of $P_o$ and $\tau_{act,p}$ the time constant for the $P_o$ change. The parameters provided by Fig. 4c indicate that at the slow AP frequency of 2.77 Hz there is a minimal $\Delta P_o(t)$ increase of 1.1% adding to the background $P_o$ of 20.5%. We conclude that mHCN2 channels contribute only minimally to the pacemaker gating, if at all. The amplitude of $P_o(t)_{2s}$ tends also to be larger for the three longer than for the shorter AP trains (Supplementary Table 5). $\langle G_{2sp}\rangle(t)$ produced a slow recovery from deactivation over many seconds (Fig. 4d) resembling that for $\langle G_{2fp}\rangle(t)$. Thus, also for the slower AP trains, mHCN2 channels are too slow to gate significantly during pacemaker depolarization.

For mHCN4V_E channels, the individual $P_o(t)_{4s}$ time courses and their average $\langle P_o(t)_{4s}\rangle$ were also time independent over the 268-ms pacemaker depolarization (Fig. 4g) and, again, the amplitude of the individual $P_o(t)_{4s}$ tend to be larger for the five longer than for the shorter AP trains (Supplementary Table 6). In analogy to mHCN2, $\langle G_{4sp}\rangle(t)$ shows a recovery from deactivation over many seconds (Fig. 4h). The stochastic switching of the channels again overlaps. As for mHCN2 channels, we see this slow recovery from deactivation as a slow relaxation process of the channel protein and this slowness is presumably the cause for the constancy of $P_o(t)_{4s}$, i.e., there is no gating during the pacemaker depolarization in mHCN4V_E channels, and presumably also in wt mHCN4 channels.

For the fast mHCN1 channels, it was intriguing to assume that at the slow AP frequency a larger time-dependent component of $P_o$ appears than in mHCN2 channels. As for the fast AP protocol, we preferred patches with fewer channels, to improve the identifiability of their switching because of their smaller conductance of 0.84 pS[13]. In the patch shown in Fig. 4i–l either 0, 1, 2, and, rarely, 3 channels were simultaneously active during pacemaker depolarization (Fig. 4j). This patch contained 3 channels. Already the individual $P_o(t)_{1s}$ time courses show pronounced activation and deactivation, despite the larger noise due to the low channel number (Fig. 4k). $\langle P_o(t)_{1s}\rangle$ indicates that on average $P_o$ varies between ~0.1 and ~0.2 during the pacemaker depolarization (Supplementary Table 7). Fitting the activation phase of $\langle P_o(t)_{1s}\rangle$ with Eq. (1) yielded that there is a $\Delta P_o(t)$ increase of 9.6% adding to the background $P_o$ of 10.0%, resulting in a 96% increase of $P_o$. Hence, mHCN1 channels are suitable to significantly contribute to the pacemaker depolarization. When plotting $\langle G_{1sp}\rangle(t)$, we observed only a stochastic fluctuation along the successive APs (Fig. 4l) as for the fast AP frequency of 10 Hz in these channels (Fig. 3l), confirming the rapid gating. These results suggest, that mHCN1 channels recover much faster from deactivation than mHCN2 and mHCN4V_E channels and are therefore relevant for pacemaking.

**Quantifying the kinetics of the HCN channels.** The slow $\langle G_{xyp}\rangle(t)$ time courses for mHCN2 and mHCN4V_E channels were fitted by an exponential function (Fig. 5a–d) yielding means for a recovery time constant $\tau_{rec}$ (Fig. 5g). These values are in the range of 8 to 15 seconds, irrespective of using the fast or slow AP frequency. These time constants are much longer than the time interval given by the reciprocal of our used AP frequencies 10 or 2.77 Hz (ochre lines in Fig. 5g).

Since for HCN1 channels the individual $\langle G_{1yp}\rangle(t)$ plots did not resolve a process of activation, we averaged 19 $\langle G_{1fp}\rangle(t)$ plots which allowed us to estimate a value for $\tau_{rec}$ of 115 ms by an exponential fit for the recovery process (Fig. 5e). In the averaged slower $\langle G_{1sp}\rangle(t)$ plot, this component was not resolved (Fig. 5f). Moreover, both plots show that mHCN1 channels are available for the pacemaker gating all the time. A slow partial inactivation process after the pulse to −30 mV (Fig. 5e, f) is not further considered herein. As a second estimate for the relaxation speed in mHCN1, we took the activation speed during pacemaker depolarization (c.f. Figure 4k), but fitted the individual $P_o(t)_{1s}$ time courses with a single exponential to obtain a measure for the variability among the experiments (for representative fits see Supplementary Fig. 7). As a result, we obtained $\tau_{act,p}$ = 43±11 ms ($n$ = 10) which is faster than the 115 ms from $\langle G_{1fp}\rangle(t)$ plot but well in the same order of magnitude (Fig. 5g).

## Discussion

Our findings, obtained with a dynamic action-potential clamp at femtosiemens resolution, show that mHCN1 channels operate at typical neuronal pacemaker potentials, whereas mHCN2 and mHCN4V_E channels, and presumably also mHCN4 wt channels, are two orders of magnitude slower. In terms of their role in pacemaking, we propose that only mHCN1 channels are relevant as pacemaker channels in the sense that they activate and deactivate during the pacemaker phase, whereas mHCN2 and mHCN4 channels contribute to pacemaking by setting the voltage level (Fig. 5h). Since these two isoforms are much more sensitive to cAMP[1,13,14,18], their role at varying cAMP levels would be to set the working point for mHCN1 channels and all other voltage-dependent channels. HCN3 was not considered herein because its activation is similarly slow or even slower than that of HCN4[12,13]. It is likely that its role is also to set the working point for mHCN1 and other channels.

Our experiments also showed that long AP trains applied to mHCN2 and mHCN4V_E channels evoked statistically larger $P_o(t)_{xy}$ values than shorter AP trains (Figs. 3c, g; 4c, g), matching the slow recovery from deactivation. Regarding the average $\langle P_o(t)_{xy}\rangle$ of 0.2 and 0.3, and even the significantly larger values in experiments with long AP trains (Figs. 3c, g; 4c, g), these values are surprisingly large if compared to results obtained by measurements of steady-state activation parameters with long rectangular pulses. This is because also long pulses usually do not lead to true steady states[40], in particular at the low voltages near the pacemaker potential. Instead, our dynamic AP clamp at single-channel resolution seems to provide physiologically more meaningful information. For a neuron expressing mHCN2 or mHCN4V_E, our results suggest that long-lasting spike activity leads to an enhanced non-specific conductance in a use-dependent manner, enabling a stronger cAMP control on the neuronal activity. Apart from this effect of the length of AP trains, our measurements showed for all isoforms marked variability of the individual $P_o(t)_{xy}$ values (Figs. 3c, g, k; 4c, g, k; Supplementary Tables 2-7). Possible reasons for this variability are the differences of the voltage shift $\Delta V$ among the patches, which we set to a mean value (see Supplementary Table 1) and the stochastic activity of single channels per se over time. Despite this variability in the amplitude of $P_o(t)_{xy}$ among the measurements, the

# AP frequency 2.77 Hz

exclusive result of a pronounced activation and deactivation gating in mHCN1 at the slow AP frequency is remarkably consistent (Fig. 4k).

Methodologically, our study was conducted at room temperature because we required *Xenopus* oocytes producing these extraordinarily tight seals[13] for femtosiemens resolution, and these oocytes do not tolerate higher temperatures. Regarding the meaning of our results for 15 degrees higher physiological temperature conditions, we assume

that they are relevant for many autorhythmic neuronal APs in the range of several tens of Hz. This is because, at the higher temperature, the basic mechanisms are presumably similar, but the steepness of the pacemaker potential is larger, generating an earlier reaching of the threshold for the activation of voltage-gated channels such as T-type $Ca^{2+}$ channels. Our experimental data show that when considering the voltage range of activation and deactivation of the HCN1 channels

**Fig. 4 | Activation of mHCN channels at the slow AP frequency of 2.77 Hz.** Three representative experiments are illustrated. Analogue to Fig. 3. The current recordings followed a 4 s pulse to −30 mV. **a**–**d** mHCN2. **a**, Current trace with one channel opening. **b**, Conductance $G(t)_{2s}$ as function of time. 42 traces are overlaid. **c**, 11 $P_o(t)_{2s}$ measurements (gray: AP trains with 50 APs; light brown: longer AP trains; black: $\langle P_{2s}\rangle(t)$ over all 11 measurements). For statistics see Supplementary Table 5. $P_o(t)_{2s}$ is constant over the time interval. The red curve is a fit of activation in $\langle P_o(t)_{2s}\rangle$ with Eq. (1) which is also shown in the inset at expanded ordinate. **d**, Averaged conductance $\langle G_{2sp}\rangle(t)$ during pacemaker depolarization. Representative experiment. Similar to the fast AP frequency, $\langle G_{2sp}\rangle(t)$ rises slowly over seconds. **e**–**h** mHCN4V_E. **e**, Current trace with two channels opening. **f** Conductance $G(t)_{4s}$

as function of time. 49 traces are overlaid. **g**, 12 $P_o(t)_{4s}$ measurements; analogue to c with Supplementary Table 6. The inset shows $\langle P_o(t)_{4s}\rangle$ at expanded ordinate. **h**, $\langle G_{4sp}\rangle(t)$ during pacemaker depolarization rises also slowly over seconds. Representative experiment. **i**–**l** mHCN1. **i** Current trace with two channels opening. **j** Conductance $G(t)_{1s}$ as function of time. 48 traces are overlaid. **k** 11 $P_o(t)_{1s}$ measurements; analogue to c with Supplementary Table 7. The activation time course was fitted in the indicated interval by Eq. (1) which is also shown in the inset at expanded ordinate (red curve). **l** Average conductance $\langle G_{1sp}\rangle(t)$ during pacemaker depolarization. Representative experiment. As for the fast AP frequency, $\langle G_{1sp}\rangle(t)$ reveals solely a stochastic activity over the 48 traces.

during pacemaker depolarization, a plateau occurs at around −72 mV (c.f. Figure 4k, middle). This finding suggests that their contribution to pacemaking is limited to the initial phase of the depolarization, as the plateau falls well below the action potential threshold.

To further explore this observation and quantitatively estimate the effects of the HCN1 isoform on pacemaking in a neuron, we simulated membrane potential dynamics in a spherical, isopotential model SCN neuron. HCN1 conductance time courses were derived from the individual experimental $P_o$ measurements (c.f. Figure 4k) to include their statistical variability (Fig. 6a, top). The model reveals that as a sole pacemaker conductance, HCN1 will only depolarize the neuron to at most ~ −73 mV (Fig. 6a, middle) across a wide range of simulation parameters, corresponding to only the initial ~25% of the total depolarization required to reach the threshold (Fig. 6b, c). To reconstruct the full depolarizing pacemaker ramp, an additional depolarizing conductance is required to take over pacemaking after this initial phase ('injected' in Fig. 6d). This suggests that HCN1 acts as a trigger channel for pacemaker depolarization, while other conductances, such as those from T-type calcium channels, are required to sustain the process. This limited contribution to pacemaking may also explain why in forebrain-wide HCN1 KO animals, gross anomalies were neither found in sleep-wake cycles nor in the EEG[42], as the loss of HCN1 currents may be compensated for by other conductances. While previous work has established that HCN1 activates faster and at more depolarized potentials than other HCN family members, our simulations constrained by experimentally measured HCN1 gating show that, under realistic biophysical conditions, dynamic HCN gating in the physiological sub-threshold range is effectively confined to HCN1, with additional HCN family conductances contributing primarily tonic background currents. In this context, our results provide a framework that may help explain why HCN1 channels are particularly suited to temporally confine sub-threshold synaptic integration, a property previously implicated in the regulation of learning and memory in CA1 pyramidal neurons[43,44].

Finally, it should be noted that HCN channels form heteromers, a phenomenon demonstrated for various isoforms, including the combinations of HCN1 with HCN2[13,14] or HCN4[24]. It is therefore likely that heteromers with either one, two, or three HCN1 subunits activate faster than homomeric HCN2 or HCN4 channels alone. This would mean that HCN1 subunits may also generate intermediately rapid pacemaker channels as heteromers. Future studies with respective HCN concatemers are promising to clarify this issue.

## Methods
### Ethical statement
All experimental procedures involving *Xenopus laevis* were approved by the Thüringer Landesamt für Verbraucherschutz (permit number UKJ-23-005) and were conducted in accordance with institutional guidelines of Friedrich Schiller University Jena and the German Animal Welfare Act.

### Oocyte preparation and cRNA injection
Oocytes were surgically obtained under anesthesia (0.3% 3-aminobenzoic acid ethyl ester) from adult females of *Xenopus laevis* at the age of 3-4 years.

The oocytes were incubated with collagenase A (3 mg/ml, Roche, Grenzach-Wyhlen, Germany) for 105 min in Ca²⁺-free Barth´s solution which containing (in mM) 82.5 NaCl, 2 KCl, 1 MgCl₂, 5 HEPES, pH 7.5. Thereafter the oocytes at stage IV and V were manually isolated. They were injected with ~50 ng cRNA transcribed from the respective coding sequences in pGEM derivatives. We used WT mHCN1 (NM_010408), mHCN2 (NM_008226), mHCN4 (NM_001081192) and its mutation mHCN4V487E[13], which is denoted herein mHCN4V_E. After cRNA injection, the oocytes were incubated at 18 °C in Barth solution containing (in mM) 84 NaCl, 1 KCl, 2.4 NaHCO₃, 0.82 MgSO₄, 0.41 CaCl₂, 0.33 Ca(NO3)₂, 7.5 TRIS, pH 7.4. The incubation times were: mHCN1: 1-2 days, mHCN2: 1-2 days, mHCN4 4-7 days and mHCN4V_E 2-4 days.

### Molecular biology
The mouse genes for mHCN1, mHCN2, and mHCN4 and the mutant mHCN4V_E were subcloned behind the T7 promoter of pGEMHEnew[13]. The Point mutation was introduced via the overlapping PCR-strategy followed by fragment subcloning by using flanking restriction sites. Sequences were checked by restriction analysis and sequencing (Microsynth SEQLAB, Göttingen, Germany). cRNAs were generated using the mMESSAGE mMACHINE T7 Kit (Thermo Fisher Scientific, Dreieich, Germany).

### Two-electrode voltage clamp
Whole-cell currents in *Xenopus* oocytes were recorded with the two-electrode voltage clamp (TEVC) technique (OC725C amplifier, Warner Instrument, Hampden, U.S.A.) at room temperature. The microelectrodes were filled with 3 M KCl. Their resistance was 0.3-1 MΩ. The cells were bathed in ND96 medium containing (in mM) 96 NaCl, 10 Hepes, 2 KCl, 1.8 CaCl₂ and 1 MgCl₂, pH 7.4. The solution was supplemented with 1 mM BaCl₂, 100 μM LaCl₃ and 100 μM GdCl₃ to minimize the impact of endogenous channels[45]. The experiments were controlled by the HEKA Patchmaster software (v2x90.5) and LIH8+8hardware (HEKA Elektronik Dr. Schulze GmbH, Lambrecht, Germany). The holding potential was generally set to −30 mV. Two types of voltage protocols were used. (1) With the standard protocol, currents were measured with prepulses between −130 and −20 mV, spaced 10 mV, followed by a test pulse to −100 mV to determine steady-state activation from the instantaneous current. The prepulse duration was 4 s except for mHCN1 where it was set to 1 s. The respective pulsing rates were 0.125 and 0.333 Hz. (2) For determining activation in the range of pacemaker depolarizations, currents were measured with a slow protocol with prepulses between −80 and −48 mV, spaced 4 mV, followed by a pulse to −100 mV. The prepulse duration of this slow protocol was generally 16 s except for mHCN1 where it was set to 4 s. The respective pulsing rates were 0.333 and 0.03125 Hz.

For mHCN4V_E, steady-state activation was determined from the amplitude of the instantaneous current at the test pulse of −100 mV, measured as mean current of the time interval between 10 to 40 ms after the begin of the test pulse. The relative amplitude of the tail current, $I/I_{max}$, was determined by relating the actual amplitude $I$ of the instantaneous current to $I_{max}$ following the prepulse to −130 mV. These values were plotted as function of voltage and fitted by the Boltzmann

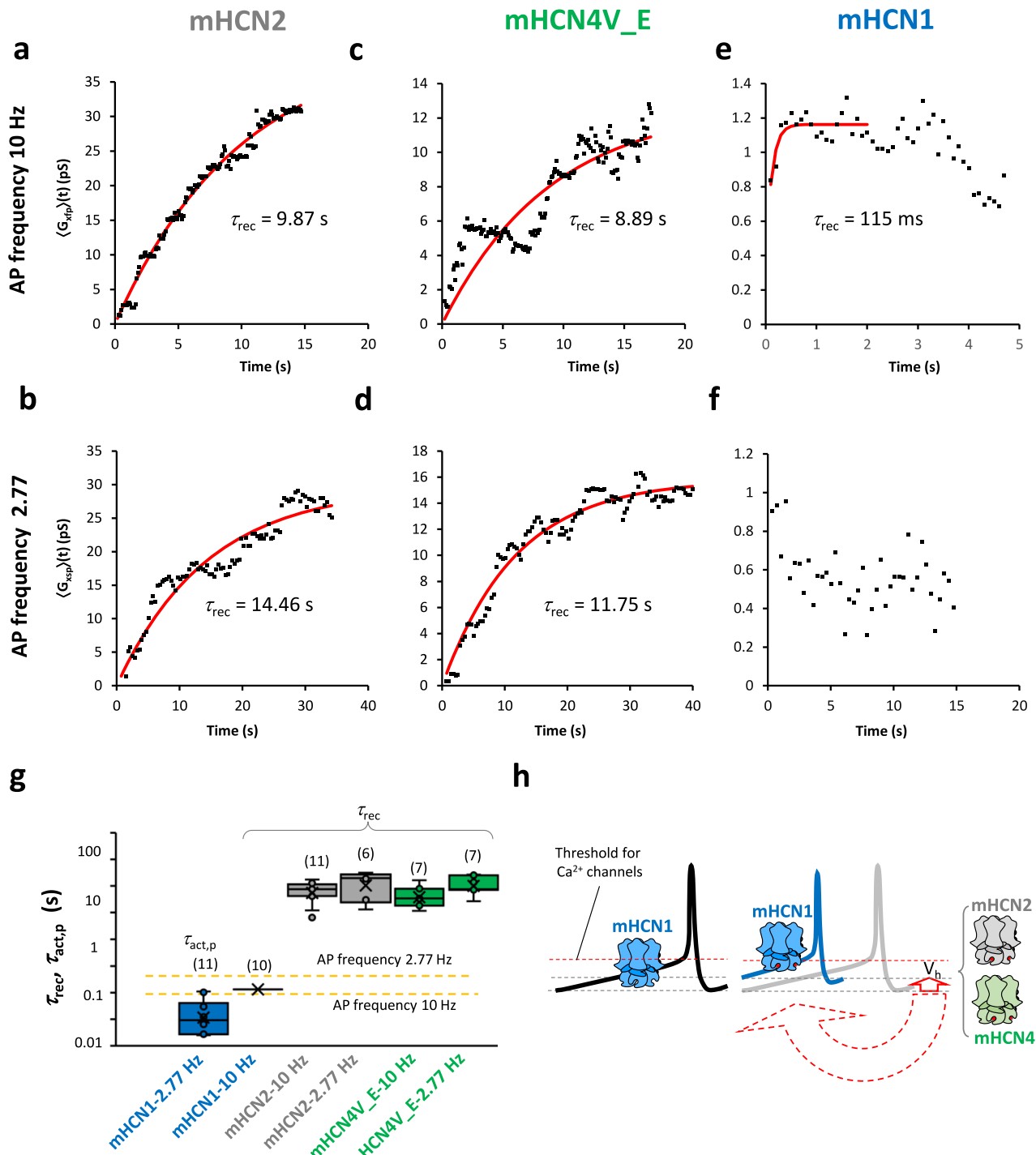

**Fig. 5 | Quantification of the relaxation kinetics in mHCN channels.** Representative examples of time courses of $\langle G_{xyp}\rangle(t)$ with the fast and the slow AP frequency fitted by an exponential function yielding the recovery time constant $\tau_{rec}$. **a** mHCN2. AP frequency 10 Hz. **b** mHCN2. AP frequency 2.77 Hz. **c** mHCN4V_E. AP frequency 10 Hz. **d** mHCN4V_E. AP frequency 2.77 Hz. **e** mHCN1. AP frequency 10 Hz. 19 $\langle G_{1fp}\rangle(t)$ plots were averaged and the initial 2 seconds were fitted. **f** mHCN1. AP frequency 2.77 Hz. 13 $\langle G_{1sp}\rangle(t)$ plots were averaged. A recovery process could not be resolved. Both $\langle G_{1yp}\rangle(t)$ plots suggest an overlapping partial inactivation process which is not further considered. **g** Box plot of the obtained relaxation time constants $\tau_{rec}$ for the three mHCN channels at the AP frequency of 10 and 2.77 Hz. Indicated are the mean (crosses), the median (horizontal line), upper and

lower quartile (box), upper and lower outliers, maximum and minimum excluding outliers (whiskers). The values were determined as described above. The number in brackets indicate the included experiments. The diagram is complemented by $\tau_{act,p}$ as a second estimate for the relaxation speed in mHCN1 (see Fig. 4k). As mean of 11 fits we obtained $\tau_{act,p} = 43 \pm 11$ ms (mean ± SEM). Notably, mHCN1 relaxes much faster than the other two isoforms. The horizontal lines indicate the times given by the reciprocal AP frequencies. **h** Schematic of the different roles of mHCN1, mHCN2, and mHCN4 channels in neuronal pacemaker activity. mHCN1 channels contribute to the pacemaker current whereas mHCN2 and mHCN4 would set the working point for mHCN1 channels, and, all other voltage-dependent channels.

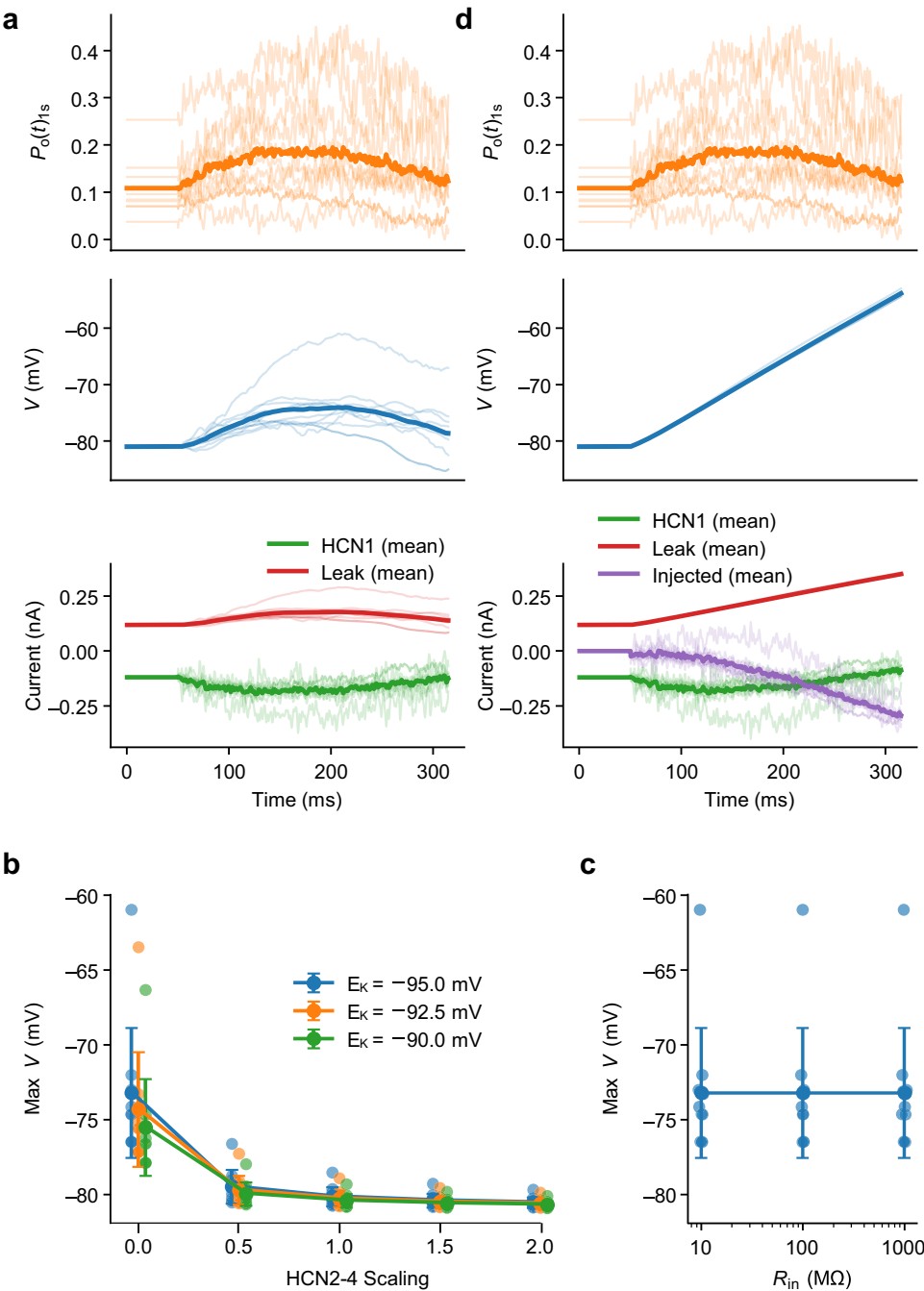

**Fig. 6 | Contribution of HCN1 to the pacemaker potential.** The time and voltage dependent conductance mediated by HCN1 was simulated in a spherical, iso-potential model SCN neuron. **a** HCN1 open probability (top), simulated membrane potential (middle), and ionic currents (bottom). The simulation begins with a 50 ms holding period followed by 11 HCN1 open probability time courses, $P_o(t)_{1s}$, measured with the slow AP clamp waveform (c.f. Figure 2c). Input resistance, leak and HCN1 conductance were constrained to yield a minimum membrane potential of −81 mV, an input resistance of 100 MΩ and a membrane time constant of 30 ms (for details see "Methods"). Traces show individual replicate simulations (transparent) and the average across $n = 11$ simulations (solid). Following the usual convention, depolarizing currents are negative. If only HCN1 is active during the pacemaker phase according to the experimentally determined $P_o(t)_{1s}$, the model neuron is on average depolarized only from −81 mV to ~ −73 mV. **b** Maximum membrane potential (Max $V$) reached during the pacemaking phase as a function of a scaling factor (multiple of the full HCN1 conductance) for an additional, constant non-selective cation conductance, representing e.g., HCN2-4, shown for three

potassium reversal potentials $E_K$. Symbols/lines indicate mean ± SD across $n = 11$ replicates; points show individual replicate values. For the simulations in panel b, $R_{in}$ was set to 100 MΩ. **c** Maximum membrane potential (Max $V$) reached during the pacemaking phase as a function of input resistance $R_{in}$ (logarithmic scale). Symbols/lines indicate mean ± SD across $n = 11$ replicates; points show individual replicate values. For the simulations in panel c, $E_K$ was set to −90 mV and the scaling factor was set to 0. Varying $R_{in}$ only rescales the conductances without changing the voltages toward which the currents drive the membrane (see "Methods"), and therefore has no effect on the maximum membrane potential reached during the pacemaking phase. **d** Contribution of HCN1 to the pacemaker potential with injected current ramp. Similar to (**a**) but with additional injection of a current to achieve the depolarizing pacemaker ramp used as a command waveform for the experimentally determined $P_o(t)_{1s}$. The bottom panel includes the HCN1 current, leak current, and the externally injected current required to maintain the ramp. For the simulations in panels a and d, $E_K$ was set to −90 mV, $R_{in}$ was set to 100 MΩ and the scaling factor was set to 0.

equation according to

$$\frac{I}{I_{max}} = \frac{1}{\left\{1 + \exp\left[\frac{z\delta_{w4m}F(V - V_{hw4m})}{RT}\right]\right\}}. \quad (2)$$

$V_{hw4m}$ and $z\delta_{w4m}$ are the half-maximum activation and the equivalent gating charge for whole oocytes for mHCNV_E. $R$ is the molar gas constant, $T$ the temperature in Kelvin (K), and $F$ the Faraday constant. For mHCN1, mHCN2 and mHCN4, the values of $V_{hwx}$ and $z\delta_{wx}$ were taken from a previous report (Supplementary Table 1)[13].

The activation speed was quantified by the time constant $\tau_{act,wf}$ and $\tau_{act,ws}$ for the fast and slow protocol, respectively. The time constants $\tau_{act,wx}$ and the delay interval $t_0$ were obtained by fitting the following function to the current time courses

$$I(t) = \begin{cases} I_{inst}, & \text{for } (t < t_0) \\ A \times \left\{1 - \exp\left[-\frac{(t-t_0)}{\tau_{act,wx}}\right] + I_{inst}\right\}, & \text{for } (t \geq t_0) \end{cases} \quad (3)$$

$A$ and $I_{inst}$ are the amplitude of the time-dependent and time-independent current component, respectively.

Fits were performed by the nonlinear least-squares fitting routines implemented in the IgorPRO 7.08 (Wavemetrics, Lake Oswege (OR), USA) and OriginPro 2016G software (OriginLab Corporation, Northampton, MA, USA).

## Patch-clamp recording at single-channel resolution

Currents at single-channel resolution were recorded with the patch-clamp technique from cell-attached patches of *Xenopus* oocytes expressing the respective channels. The K$^+$ rich bath solution was assumed to zero the resting potential of the oocytes. All recordings were performed at room temperature (20–22 °C). The patch pipettes were pulled from thick-walled quartz tubing (outer and inner diameter 1.0 and 0.5 mm, respectively) using the P-2000 puller, Sutter Instruments, Novato (CA), USA, to keep the RC and dielectric noise as low as possible[46]. The resistance of the pipettes was 7 to 14 MΩ. The bath and pipette solution contained (in mM) 100 KCl, 10 EGTA, 1 MgCl$_2$, 10 Hepes (pH 7.2) and 120 KCl, 10 Hepes, 1 MgCl$_2$, 1 CaCl$_2$ (pH 7.2) respectively.

Particular care was taken to shield the patches, pipette and bath. Most recordings were performed by using an additional small Faraday box manufactured from V2A steel. This additional Faraday box was positioned on the stage of the inverted microscope. The microscope, headstage of the amplifier and micromanipulator (MP-285, Sutter, U.S.A) were surrounded by a large Faraday box (cage-in-cage strategy). The central ground was connected to the amplifier ground whereas the chamber, the small Faraday box and the micromanipulator were grounded via the headstage ground. With this shielding we could reliably remove all 50 Hz components. Occasional spikes, presumably by inductive sources, could not be completely removed but were sufficiently rare to exclude these traces if required.

In part of the experiments with the smaller unitary currents of mHCN1 we use a modified version of shielding: Around the inner small Faraday box we positioned an extra small Faraday box manufactured from thick-walled ferroelectric iron that was also connected to the headstage ground. Our impression was that this extra box improved the total shielding to some extent.

Currents were recorded with an Axopatch 200B amplifier (Axon Instruments Inc., Foster City (CA), USA) in the capacitive mode. Stimulation and data recording were performed with the ISO3 hard- and software (MFK, Niederhausen, Germany). The sampling rate was 20 kHz for the faster mHCN1 channels and 5 kHz for the other channels. The on-line filter of the amplifier (4-pole Bessel) was set to 1 kHz.

Reasonable patches had a seal resistance of >100 GΩ, in many cases 200 to 700 GΩ, or exceptionally also higher. An excellent seal resistance was a prerequisite to obtain sufficiently low noise and stable conditions for our analyses. Our method with this advanced resolution was described in a previous study[13]. Based on a previous study on single HCN2 channels expressed in *Xenopus* oocytes[47], ruling out cooperativity between the channels, we assumed that all channels studied herein act as independent molecules.

## Single-channel analysis

The recorded data, on-line filtered at 1 kHz, were off-line filtered with a digital Gauss-filter down to either 200 Hz, 100 Hz or 50 Hz by the ISO3 software. The channel number in a patch, $N$, was obtained from the highest number of unitary steps within a trace at −160 mV or, rarely, at −130 mV if 10 or less channels were present, assuming that at these voltages all channels are open. If the currents were larger due to more channels in the patch, the channel number was estimated at the same voltages by dividing the amplitude of the late current at these voltages by the means of the respective single-channel currents which were for −160 mV: mHCN1, −126±6 fA ($n = 9$); mHCN2, −246±4 fA ($n = 10$); mHCN4V_E, −181±4 fA ($n = 16$). The respective values for −130 mV were: mHCN1 −115±4 fA ($n = 28$), mHCN2 −220±4 fA ($n = 22$), mHCN4V_E −171±4 fA ($n = 25$). As performed previously, the amplitudes of the unitary currents were determined locally by individual openings and closures[13].

## Ensemble currents in cell-attached patches and Boltzmann function

Steady-state activation was also determined for cell-attached patches to quantify the voltage shift compared to the whole-cell recordings with the TEVC technique. The data points were obtained with the short pulse protocol. $I/I_{max}$ of the instantaneous tail current was plotted as function of voltage accordingly and fitted with Eq. (2) yielding for the patches the parameters $V_{hpx}$ and $z\delta_{px}$ instead of $V_{hwx}$ and $z\delta_{wx}$.

## AP clamp in cell-attached patches at single-channel resolution

An action potential clamp was performed with trains of idealized action potentials. To design an idealized action potential, a published action potential of a suprachiasmatic nucleus cell recorded at room temperature was used as template (Fig. 2a)[31]. The idealized action potential was obtained by a sequence of five voltage ramps resembling the reported time course (ochre trace in Fig. 2b). This AP was termed here as fast protocol AP$(t)_{xf}$. The resulting AP frequency was 10 Hz. The voltages for AP$(t)_{xy}$ were corrected by a voltage shift $\Delta V_x$ because steady-state activation in the cell-attached patches was more negative with respect to the TEVC data by unknown reasons. 'x' denotes the channel type (x = 1,2,4) and 'y' is either 'f' or 's' for the fast (10 Hz) and slow (2.77 Hz) AP-protocol. To quantify $\Delta V_x$, we determined steady-state activation as performed in the TEVC measurements (Supplementary Fig. 3). The values for $\Delta V_x = V_{hpx} - V_{hwx}$ were closely similar for the three channels: −26 mV (mHCN1), −28 mV (mHCN2), −25 mV (mHCN4V_E) (Supplementary Table 1), resulting in the command potentials shown in Fig. 2b.

Because our data acquisition system required per action potential 20 ms for storage on the hard disk, in which recoding was not possible, we set the voltage after repolarization to the maximum diastolic potential for 20 ms before starting the next pacemaker depolarization. This command signal was on-line filtered at 400 Hz (8-pole Bessel) to reduce capacitive artefacts when switching between two ramps. The resulting AP$(t)_{1f}$, AP$(t)_{2f}$ and AP$(t)_{4f}$ differ solely by the slightly different $\Delta V_x$ values (Fig. 2b). We applied trains containing at least 50 AP$(t)_{x,y}$ and each train was preceded by a 4 s pulse to −30 mV to deactivate the channels. After the end of an AP train another 4 s pulse to −30 mV was applied.

In order to test the influence of the AP frequency, and thus the length of the pacemaker depolarization, on the performance of the channels, we used also the slower AP frequency of 2.77 Hz (Fig. 2c).

This command signal was on-line filtered at 100 Hz (8-pole Bessel). This resulted in $AP(t)_{1s}$, $AP(t)_{2s}$ and $AP(t)_{4s}$. Again, we applied trains of at least 50 $AP(t)_{x,y}$ and each train was preceded by a 4 s pulse to −30 mV to deactivate the channels.

The current signals $I(t)$ were filtered off-line down to either 200 Hz, 100 Hz or 50 Hz.

## Calculating $G(t)_{xy}$ and $P_o(t)_{xy}$

Because our resolution allowed us to identify single-channel activity in the pacemaker range, it became also possible to identify traces with minimum inward current and absent single-channel activity. These traces were declared as 'nulls'. In our search for nulls, we noticed for mHCN2 and mHCN4V_E channels a pronounced slow increase of the current at pacemaker potentials during the AP train following the 4 s pulse to −30 mV. The best chance for obtaining a null was thus given by the first AP in the train. Available nulls in a patch were identified selected and averaged. The average was subtracted from all current traces of the train, yielding the current time courses, $I(t)_{xy}$. The $I(t)_{xy}$ traces were essentially free of leak and capacitive current components. Because in $I(t)_{xy}$ the voltage, and thus also the driving force, changes along the clamped $AP(t)_{xy}$, the conductance $G(t)_{xy}$ in the patch was calculated by

$$G(t)_{xy} = I(t)_{xy}/AP(t)_{xy} \tag{4}$$

The analysis was confined to the pacemaker depolarization only because during the steep phases of the action potential Eq. (4) could produce unrealistic values and also division by zero was avoided.

If the activity of 20 or less channels overlapped and the noise was sufficiently low, the $G(t)_{xy}$ plot resolved the action of single channels, which is facilitated by the relatively long open times of HCN channels and the factual absence of open-channel noise[13]. The $G(t)_{xy}$ plot during the pacemaker depolarization provides the unique advantage that the channel nature of the current can be continuously examined without using any pharmacological intervention.

From $G(t)_{xy}$ we calculated $P_o(t)_{xy}$ by

$$P_o(t)_{xy} = G(t)_{xy}/G_{sat} \tag{5}$$

where $G_{sat}$ was obtained by $G_{sat}=I_{sat}/V_{sat}$ in each patch. $V_{sat}$ is the used saturating voltage of −160 mV (or rarely of −130 mV for mHCN1) and $I_{sat}$ the corresponding current amplitude at the end of the pulse to $V_{sat}$. Notably, $G(t)_{xy}$ is given by the product $N\gamma P_o(t)_{xy}$ where N is the number of active channels in the patch and γ the single-channel conductance. For a given patch a change of $G(t)_{xy}$ therefore reflect a change of $P_o(t)_{xy}$.

## Statistics and fits of single-channel data

If not otherwise noted the data are given as mean±SEM. An AP train following a 4s-pulse to −30 mV consisted in most cases of 50 APs or longer if indicated. The currents measured for an AP train were treated as independent sample. Up to three such measurements were conducted on a patch. Since traces with clear disturbing interferences were eliminated manually, the actual number of AP currents included in the analysis could be below the number of the applied APs. The average $\langle P_o(t)_{xy}\rangle$ contained data from 5 to 8 patches. For statistical details see Supplementary Tables 2-7. The activation phase in the pacemaker depolarization of both the individual $P_o(t)_{1s}$ time courses as well as the average $\langle P_o(t)_{1s}\rangle$ and $\langle P_o(t)_{2s}\rangle$ time courses were fitted by Eq. (1). The recovery kinetics from deactivation were fitted by a standard exponential function.

Basic data processing was performed either with Excel (Microsoft Office 16 LTCS) or Matlab (R2018a).

## Construction of null traces from voltage-shifted AP recordings

For mHCN1 channels, true null traces were often not found. To construct null traces we recorded after the AP trains a series of AP current traces in which the voltage was shifted by $V_{shift}$ = +30 or +50 mV to deactivate all channels ($AP(t)_{Vshift}$). At least 10 of $AP(t)_{Vshift}$ traces were averaged yielding the mean $avAP(t)_{Vshift}$.

We then defined for $avAP(t)_{Vshift}$ the begin of the linear pacemaker depolarization the voltage $V_1$ and for the end the voltage $V_2$ and determined the corresponding currents $I_1$ and $I_2$, respectively. Because our pacemaker depolarization was caused by a straight line, we could assume linearity between $V_1$ and $V_2$.

Using the two-point equation for this line

$$(I - I_1)/(V - V_1) = (I_2 - I_1)/(V_2 - V_1) \tag{6}$$

we determined the slope m and the y-intercept $I_0$ according to

$$m = (I_2 - I_1)/(V_2 - V_1) \tag{7}$$

and

$$I_0 = (I_1 V_2 - I_2 V_1)/(V_2 - V_1) \tag{8}$$

yielding the equation

$$I = m \cdot V + I_0 \tag{9}$$

The corrected current for the null subtraction was determined by

$$I_{corr} = m \cdot (V_1 - V_{shift}) + I_0 \tag{10}$$

The $avAP(t)_{Vshift}$ traces were then shifted by $I_{corr}$ and could be used as null for subtraction from the individual $AP(t)$ traces. The procedure assumes linearity in the considered voltage range according to Ohm's law.

## Simulations of the effects of HCN1 channels in an SCN neuron

Simulations were performed using the Brian2 simulation environment[48]. A single-compartment spherical neuron model was implemented with a leak conductance and with non-selective cation conductances. One of the non-selective cation conductances ($g_{HCN1}$) represented HCN1. Simulations were run separately for each of the 11 $P_o(t)_{1s}$ replicates. For each replicate, the resting HCN1 open probability $P_{o,rest}$ was estimated from the initial segment of the corresponding experimental trace. An additional constant non-selective cation conductance $g_{HCNX}$ was included to represent the approximately constant open probabilities of HCN2-4 in accordance with their slow recovery from deactivation (c.f. Figure 5a–d). This conductance was varied by multiplying $g_{HCN1}$ with different scaling factors $s_{HCNX}$, i.e., $g_{HCNX} = g_{HCN1} s_{HCNX}$ (Fig. 6b).

Membrane potential dynamics were described by:

$$C_{cell} \frac{dV}{dt} = I_{leak} + I_{HCN1} + I_{HCNX} + I_{inj}, \tag{11}$$

with

$$I_{leak} = -g_{leak}(V - E_K),$$

$$I_{HCN1} = -g_{HCN1} P_o(t)(V - E_{HCN}),$$

and

$$I_{HCNX} = -g_{HCNX}(V - E_{HCN}) = -g_{HCN1} s_{HCNX}(V - E_{HCN}).$$

$E_{HCN}$ was 0 mV and $E_K$, the potassium reversal potential, was varied between −95 mV and −90 mV as indicated. At the minimum membrane potential of −81 mV, corresponding to the minimum of the experimentally determined membrane potential during pacemaking in an SCN neuron (Fig. 6), the system was assumed to be in a steady state (d$V$/d$t$ = 0), yielding the following steady-state current balance condition at −81 mV:

$$g_{leak}(V_{rest} - E_K) + (P_{o,rest} + s_{HCNX})g_{HCN1}(V_{rest} - E_{HCN}) = 0. \tag{12}$$

The total membrane conductance was constrained by the desired input resistance as follows:

$$G_{total} = 1/R_{in} = g_{leak} + P_{o,rest}g_{HCN1} + g_{HCNX} = g_{leak} + (P_{o,rest} + s_{HCNX})g_{HCN1}. \tag{13}$$

Solving Eq. (12) for $g_{leak}$ and inserting the result in Eq. (13) uniquely determined the leak conductance $g_{leak}$, the full HCN1 conductance $g_{HCN1}$, and the additional constant non-selective cation conductance $g_{HCNX}$ for each parameter combination. The input resistance $R_{in}$ was varied between 10 MΩ and 1 GΩ, as indicated in Fig. 6c, covering a realistic range for SCN neurons[49]. The membrane time constant $\tau_m = R_{in} C_{cell}$ was set to 30 ms[49], which allowed us to determine a unique value for the whole-cell capacitance $C_{cell} = \tau_m/R_{in}$. Simulations were performed with a fixed time step of 10 μs.

Simulations without and with an externally injected current were run to assess the impact of HCN dynamics on membrane potential (Fig. 6d). The injected current was computed to produce a linear depolarizing voltage ramp from −81 mV to −53 mV over a duration of 268 ms, thereby yielding the depolarizing pacemaker ramp used as a command waveform for the experimentally determined $P_o(t)_{1s}$.

## Reporting summary
Further information on research design is available in the Nature Portfolio Reporting Summary linked to this article.

## Data availability
All data included in the manuscript are deposited in the Open Science Framework [osf.io/7g5ch].

## Code availability
The code of the simulations (HCN_rest-singlePo.py) is available at OSF [osf.io/7g5ch]. No manuscript-specific software was used for the analysis of measured data.

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

## Acknowledgements

We like to thank U.S., C.R., H.G., P.H., M.B. and M.H. for the excellent technical support. This work was supported by the Deutsche Forschungsgemeinschaft through the Research Unit 2518 DynIon, project P02, to K.B.

## Author contributions

U.E. performed most TEVC measurements and their analysis. A.S. and D.T. contributed to the TEVC measurements. R.S. provided analytical tools, and C.S. conducted the molecular biology. C.S.-H. designed and performed the simulations on the SCN neuron. K.B. designed the study, did all single-channel recordings and their analysis, and wrote the manuscript.

## Funding

## Competing interests

The authors declare no competing interests.
