## [Transparent Peer Review File · Nature Communications]

HCN1 is the Sole HCN Pacemaker Channel in Neurons

Corresponding Author: Professor Klaus Benndorf

Version 0:

Reviewer comments:

Reviewer #1

(Remarks to the Author)

This manuscript by Enke et al. analyzes the dynamics of HCN1, HCN2 and HCN4 channels at femtoSiemens resolution using dynamic action potential clamp to determine the potential contributions of these channels to neuronal pacemaking. The experiments are performed on channels expressed in oocytes. The authors conclude that only HCN1 channels can participate in pacemaking activity, as the kinetics of HCN2 and HCN4 are too slow.

The experiments are technically demanding and the results support some of the conclusions drawn. The results could be potentially interesting, however, I have some major concerns, as detailed below, especially about the relevance of the findings in terms of neuronal pacemaking frequency.

Major

- 1) The authors should explain more clearly why the mHCN4V_E mutant version was used in the experiments rather than mHCN4, besides the fact that its conductance resembles that of mHCN2.
- 2) Results, Figures 2 and 3. How does the number of channels in the patch affect the results? It is mentioned in lines 166-168 that patches with fewer channels were preferred for HCN1 channels. Is it possible that having too many channels in the patches, as for HCN2 and HCN 4, confounds the findings?
- 3) Fig. 2k, lines 137-138. It is unclear what the authors mean with "a decent peak", as I do not see one in this panel. Also, the term decent seem somewhat colloquial for a scientific report.
- 4) Discussion lines 208-213: whereas I understand that the experiments cannot be performed at physiological temperature, the extrapolation of the firing frequency between 21 C and 37 C cannot be made using the Q10 of just one type of channel. Besides, for this calculation they should not be using the Q10 of the channel they are trying to demonstrate is important for pacemaking. In the same way, stretching the AP command trace to slow its frequency (extended data Fig. 7) can distort the results.

Minor

- 1) The general organization of the manuscript is somewhat confusing. Some of the extended data figures (Fig. 3, Fig. 7) should be added to the main figures as they are essential in the flow of the manuscript.
- 2) Introduction, line 48: the sentence "and it varies..." is unclear and should be rewritten.
- 3) Results line 85 there is a typo.
- 4) Results lines 165-166: it is unclear what this sentence means.
- 5) Results lines 165-166: this sentence is unclear and should be rephrased.
- 6) Discussion lines 203-205: it is unclear what sympathetic stimulation refers to in the context.
- 7) Discussion lines 228-231: CA1 neurons do not display pacemaking activity, therefore it is unclear how their role in learning and memory could be explained by the role of heteromers on pacemaking frequency.
- 8) Methods, line 327: this sentence is unclear.

Reviewer #3

(Remarks to the Author)

This paper investigates which hyperpolarization-activated cyclic nucleotide-gated (HCN) channel isoforms function as neuronal pacemaker channels during rhythmic firing. Using a dynamic clamp, the authors directly track the activity of HCN1,

HCN2, and HCN4 channels during realistic pacemaker depolarizations. They show that although all three isoforms exhibit some open probability, only HCN1 gates rapidly enough to activate and deactivate within individual action potential cycles. In contrast, HCN2 and HCN4 display gating kinetics that are too slow, contributing mainly to setting the background membrane potential rather than pacing. Simulations indicate that HCN1 provides only the initial portion of the pacemaker depolarization, acting as a trigger channel before other conductances take over. The study concludes that HCN1 is the sole genuine neuronal HCN pacemaker channel, potentially redefining the functional division of labor among HCN isoforms in the brain.

The work appears to be well executed, and the highly technical paper is well written and clear. I focus my review on the simulation component, which is the area I am most familiar with.

Overall, the simulations are done at a good standard and provide a useful component of the study. I would suggest that the authors move the supplementary figures about simulations to the main body of the paper and provide a short description of what they did and observed in Results (currently it is only mentioned briefly in Discussion). With only 4 main figures in the current version, these additions should fit easily.

In terms of the simulation detail, my main concern is about using the open probability trace averaged over 12 experiments. I would recommend that the authors repeat simulations for non-averaged data from each of the 12 experiments. Fig. 3k suggests that non-averaged $P_0(t)_{s1}$ fluctuates much more and potentially reaches higher values than the averaged version. This may affect the results. It is therefore useful to use the non-averaged $P_0(t)_{s1}$ traces. It is quite likely that the main conclusions would still hold, but it would be good to check.

For the results shown in panels b, c, d of Extended Data Figure 9, it would be best to cover the whole grid of options, rather than varying one parameter at a time while holding all the others constant. With 3 parameters used here and 3 values for the first, 3 for the second, and 4 for the third, this would result in $3 \times 3 \times 4 = 36$ conditions. For each of them, it would be best to use 12 separate non-averaged $P_0(t)_{s1}$ traces, as I suggested above. That becomes $36 \times 12 = 432$ simulations. Given that all the code exists, and the simulations of a single-compartment neuron are not computationally demanding, this should be feasible without too much trouble. Having such a more complete set of simulations would provide a more convincing evidence that the main conclusions are supported (assuming the results do not change substantially).

Version 1:

Reviewer comments:

Reviewer #1

(Remarks to the Author)

The authors addressed the issues I raised in my previous review and clarified my concerns.

Reviewer #3

(Remarks to the Author)

The authors addressed all my comments, and I have no further concerns.

REVIEWER COMMENTS

Reviewer #1 (Remarks to the Author):

This manuscript by Enke et al. analyzes the dynamics of HCN1, HCN2 and HCN4 channels at femtoSiemens resolution using dynamic action potential clamp to determine the potential contributions of these channels to neuronal pacemaking. The experiments are performed on channels expressed in oocytes. The authors conclude that only HCN1 channels can participate in pacemaking activity, as the kinetics of HCN2 and HCN4 are too slow.

The experiments are technically demanding and the results support some of the conclusions drawn. The results could be potentially interesting, however, I have some major concerns, as detailed below, especially about the relevance of the findings in terms of neuronal pacemaking frequency.

We thank Reviewer #1 for the numerous supporting suggestions and recommendations. And we also like to thank her/him for appreciating our work. We could address all four major and eight minor points adequately (see point-by-point responses below).

The most relevant changes are:

Following the suggestions, we moved the two schemes of the used command action potentials (old Supplementary Figs. 3 and 7) to the new Fig. 2 in the main manuscript.

Another change results from the suggestion of Reviewer #3, to shift the simulations (old Supplementary Figures 9 and 10) to the main manuscript to the new Fig. 6 and to repeat the analysis for the individual P_o time courses instead of using the mean P_o time course with the idea to include effects of the variability among the measurements on the results.

Following this logic and the requirements of *Nature Communications* on statistics, to present individual data underlying averages and means, we inserted in Fig. 3 and 4 c,g,k (old Fig. 2 and 3 c,g,k) the P_o time courses of the individual measurements for all three channels and the two AP frequencies. In the course of this revision and the preparation of the required Excel files, we rechecked the whole analysis of the P_o time courses, in particular for the risk of subtle leakage changes during the AP trains, which led to some altered P_o values. Most importantly, the major result of a relevant activation and deactivation only of mHCN1 at the slow AP frequency remains and is even more clear now (Fig. 4k). The now presented individual P_o time courses underscore this main result beautifully.

To make the statistics underlying the calculation of the $P_o(t)_{xy}$ time courses more transparent, we added the Supplementary Tables 2-7 containing the trace numbers of all measurements. These Supplementary Tables demonstrate that the trace numbers are large, adding up to 5479 traces for the three channels and the two AP frequencies.

The new presentation of the individual P_o time courses with their relatively large scatter in amplitude prompted us to discuss this issue. We therefore included in the Discussion the following paragraph:

Our experiments also showed that long AP trains applied to mHCN2 and mHCN4V_E channels evoked statistically larger $P_o(t)_{xy}$ values than shorter AP trains (Fig. 3c,g; Fig. 4c,g), matching the slow recovery from deactivation. Regarding the average $\langle P_o(t)_{xy} \rangle$ of 0.2 and 0.3, and even the significantly larger

values in experiments with long AP trains (Fig. 3c,g; Fig. 4c,g), these values are surprisingly large if compared to results obtained by measurements of steady-state activation parameters with long rectangular pulses. This is because also long pulses usually do not lead to true steady states⁴⁰, in particular at the low voltages near the pacemaker potential. Instead, our dynamic AP clamp at single-channel resolution seems to provide physiologically more meaningful information. For a neuron expressing mHCN2 or mHCN4V_E, our results suggest that long-lasting spike activity leads to an enhanced non-specific conductance in a use-dependent manner, enabling a stronger cAMP control on the neuronal activity. Apart from this effect of the length of AP trains, our measurements showed for all isoforms marked variability of the individual $P_o(t)_{xy}$ values (Fig. 3c,g,k; Fig. 4c,g,k; Supplementary Tables 2-7). Possible reasons for this variability are the differences of the voltage shift ΔV among the patches, which we set to a mean value (see Supplementary Table 1) and the stochastic activity of single channels per se over time. Despite this variability in the amplitude of $P_o(t)_{xy}$ among the measurements, the exclusive result of a pronounced activation and deactivation gating in mHCN1 at the slow AP frequency is remarkably consistent (Fig. 4k).

Another change is that closer inspection of the averaged P_o time course for mHCN2 at the slow AP frequency yielded a very small but still identifiable activation of some more than 1% (new Fig. 3c). Hence, there is a tiny, but certainly irrelevant, activation gating also in mHCN2 which however, does not alter the conclusions of the story.

All respective average P_o -time courses can be reconstructed from the uploaded Excel files.

Two representative fits of the activation phase in individual P_o time courses of mHCN1, as required for the mean in Fig. 5g, are shown in the new Supplementary Fig. 7.

To provide more information on data handling, we inserted in the methods part a short section 'Statistics and fits of single-channel data'.

Point-by-point responses to Reviewer #1:

Major

1)The authors should explain more clearly why the mHCN4V_E mutant version was used in the experiments rather than mHCN4, besides the fact that its conductance resembles that of mHCN2.

We used the mutant mHCN4V_E instead of mHCN4 because recording of single-channel activity is the core of the experimental part of the study and the amplitude of the single-channel currents mHCN4V_E is about 2.5 times larger than the really tiny one of wt mHCN4 (Benndorf et al., PNAS, 2025, PMID: 39879240). In contrast to the conductance, the gating of mHCN4V_E and wt mHCN4 are very similar, as demonstrated in Supplementary Fig. 2.

To emphasize this in the manuscript, we inserted the following sentence:

The use of this mutant uniquely allowed us to identify the unitary currents of mHCN4 channel gating at pacemaker voltages.

2)Results, Figures 2 and 3. How does the number of channels in the patch affect the results? It is mentioned in lines 166-168 that patches with fewer channels were preferred for HCN1 channels. Is it possible that having too many channels in the patches, as for HCN2 and HCN 4, confounds the findings?

The reviewer raises the question whether or not there is a relevant interaction of channels in our measurements, which should be more relevant with increasing channel numbers in the

patch. In other words, this addresses the question if there is cooperativity **between channels** in the activation gating.

While it is clear from various studies that there is a pronounced cooperativity **between the subunits** within a channel, including cooperativity at the level of the binding domains (PMID: 38377199), there are two controversial results for cooperativity between HCN2 channels: On the one hand, Dekker and Yellen reported cooperativity between single HCN2 channels expressed in HEK cells (PMID: 17043149). On the other hand, we ruled out in a previous report such a cooperativity between single HCN2 channels expressed in *Xenopus* oocytes (PMID: 24094399), as used herein. We therefore assume that under the experimental conditions of the present study there is no relevant cooperativity between the channels and, thus, that the channel number in a patch is not relevant.

To clarify this aspect, we inserted the following sentence in the Materials and Methods part of manuscript:

Based on a previous study on single HCN2 channels expressed in Xenopus oocytes (Thon et al., BJ, PMID: 24094399), ruling out cooperativity between the channels, we assumed that all channels studied herein act as independent molecules.

3)Fig. 2k, lines 137-138. It is unclear what the authors mean with “a decent peak”, as I do not see one in this panel. Also, the term decent seem somewhat colloquial for a scientific report.

We like to thank the reviewer for raising this point. We took it to recheck the averaged P_o time courses $\langle P_o(t)_{xy} \rangle$ for all three channels at the two AP frequencies (see above). In fact, we noticed that the ‘decent peak in the old Fig. 2k (new Fig. 3k) does not hold. After our revision, the result is that for the AP frequency of 10 Hz the P_o time course over the considered 62 ms is de facto flat. This frequency is obviously too fast for a relevant gating.

When checking the time courses of the averaged open probability with the slow AP frequency of 2.77 Hz ($\langle P_o(t)_{xs} \rangle$), we noticed that $\langle P_o(t)_{2s} \rangle$ showed also a time-dependent increase during the pacemaker interval of 270 ms, though its amplitude was really tiny, but visible at expanded ordinate (new Fig. 4c). We fitted this time course by the new equation (1) and indicate the fit parameters in the new Fig. 4c.

We now also show in Fig. 3c,g and Fig. 4c,g that longer AP trains generate larger $P_o(t)_{xy}$ values in mHCN2 and mHCN4V_E which matches the slow recovery from deactivation in these channels. The respective new Supplementary Tables 2,3,5, and 6 provide the exact statistics for the patches.

In Fig. 4k we now show, in line with the other $\langle P_o(t)_{xy} \rangle$, also the averaged $\langle P_o(t)_{1s} \rangle$ with an exponential fit, instead of a single experiment with fitted trace. The processes of activation and deactivation are also visible in the individual P_o time courses.

Two representative individual $P_o(t)_{1s}$ traces with corresponding fits are shown now in Supplementary Fig. 7.

4)Discussion lines 208-213: whereas I understand that the experiments cannot be performed at physiological temperature, the extrapolation of the firing frequency between 21°C and 37°C cannot be made using the Q10 of just one type of channel. Besides, for this calculation they should not be using the Q10 of the channel they are trying to demonstrate is important for pacemaking.

We followed the suggestion of the reviewer and took out all predictions based on Q_{10} values. Instead, the following sentence has been included in the Discussion part:

Regarding the meaning of our results for 15 degrees higher physiological temperature conditions, we assume that they are relevant for many autorhythmic neuronal APs in the range of several tens of Hz. This is because, at the higher temperature, the basic mechanisms are presumably similar, but the steepness of the pacemaker potential is larger, generating an earlier reaching of the threshold for the activation of voltage-gated channels such as T-type Ca^{2+} channels.

In the same way, stretching the AP command trace to slow its frequency (extended data Fig. 7) can distort the results.

We agree that our way of stretching the APs is arbitrary to some extent. However, we think that this simple approach is useful to study the channels at lower AP frequencies. In fact, the biological variety of autorhythmic action potentials is large and our stretched AP falls within this variety. In addition, we inserted in the Results part a subclause with a reference for a neuronal action potential recorded at room temperature which resembles the shape of our stretched action potential.

The subclause reads:

...a frequency that is also well in the range of autorhythmic firing at room temperature, as e.g. in rat cerebellar Golgi cells (Forti et al., 2006, PMID: 16690702).

Minor

1)The general organization of the manuscript is somewhat confusing. Some of the extended data figures (Fig. 3, Fig. 7) should be added to the main figures as they are essential in the flow of the manuscript.

We followed the suggestion of the Reviewer and put the old Supplementary Figures 3 and 7 into the new Figure 2 of the main manuscript.

2)Introduction, line 48: the sentence “and it varies...” is unclear and should be rewritten.

We revised the sentence and emphasize that differences exist between the results of different reports. It reads now:

... but shows substantial variation for each isoform across different studies.

3)Results line 85 there is a typo.

Thanks. We corrected mHCN1V_E to mHCN4V_E.

4)Results lines 165-166: it is unclear what this sentence means.

In the context that we include now a tiny change of $\langle P_o(t)_{2s} \rangle$, we changed the sentence to:

For the fast mHCN1 channels, it was intriguing to assume that at the slow AP frequency a larger time-dependent component of P_o appears than in mHCN2 channels.

5)Results lines 165-166: this sentence is unclear and should be rephrased.

See previous point.

6)Discussion lines 203-205: it is unclear what sympathetic stimulation refers to in the context.

Thanks for this criticism. We revised the sentence and replaced 'sympathetic stimulation' by 'changed cAMP levels'.

The sentence reads now:

Since these two isoforms are much more sensitive to cAMP (References), their role at varying cAMP levels would be to set the working point for mHCN1 channels and all other voltage-dependent channels.

7) Discussion lines 228-231: CA1 neurons do not display pacemaking activity, therefore it is unclear how their role in learning and memory could be explained by the role of heteromers on pacemaking frequency.

Thanks. We have clarified this point in the revised discussion:

While previous work has established that HCN1 activates faster and at more depolarized potentials than other HCN family members, our simulations constrained by experimentally measured HCN1 gating show that, under realistic biophysical conditions, dynamic HCN gating in the physiological subthreshold range is effectively confined to HCN1, with additional HCN family conductances contributing primarily tonic background currents. In this context, our results provide a framework that may help explain why HCN1 channels are particularly suited to temporally confine subthreshold synaptic integration, a property previously implicated in the regulation of learning and memory in CA1 pyramidal neurons (References).

8) Methods, line 327: this sentence is unclear.

We removed the sentence because it seems to be clear that we used our AP protocols which are positioned more prominently now in the new Fig. 3.

Reviewer #3 (Remarks to the Author):

We like to thank Reviewer #3 for the appreciation of our work, and, in particular, for the supportive criticisms on our simulations of the consequences of our results for pacemaking in neurons. We followed her/his suggestion to move the old Supplementary Figures 9 and 10 to the new Fig. 6 in the main manuscript. Moreover, we included the P_o time courses of the individual measurements in the new Fig. 6 (see point-by-point responses below).

The most relevant changes are:

Following this logic and the requirements of *Nature Communications* on statistics, we now inserted in Fig. 3 c,g,k and Fig. 4 c,g,k (old Fig. 2 and 3 c,g,k) also the P_o time courses of the individual measurements for all three channels and the two AP frequencies. In the course of the revision and the preparation of the required Excel files, we checked the whole analysis of the P_o time courses, in particular for the risk of subtle leakage changes during the AP trains, which led to slightly altered P_o values. We also removed a scaling error. Most importantly, the major result of a relevant activation and deactivation only of mHCN1 at the slow AP frequency remains and is even more clear now (Fig. 4k). The individual time courses underscore this main result beautifully.

To make the statistics underlying the calculation of the $P_o(t)_{xy}$ time courses more transparent, we added the Supplementary Tables 2-7 containing the trace numbers of all measurements. These Supplementary Tables demonstrate that the trace numbers are large, adding up to 5479 traces for the three channels and the two AP frequencies.

We now also show in Fig. 3c,g and Fig. 4c,g that longer AP trains generate larger $P_o(t)_{xy}$ values in mHCN2 and mHCN4V_E which matches the slow recovery from deactivation in these channels.

The new presentation of the individual P_o time courses with their relatively large scatter in amplitude prompted us to discuss this issue. We therefore included in the discussion the following paragraph:

Our experiments also showed that long AP trains applied to mHCN2 and mHCN4V_E channels evoked statistically larger $P_o(t)_{xy}$ values than shorter AP trains (Fig. 3c,g; Fig. 4c,g), matching the slow recovery from deactivation. Regarding the average $\langle P_o(t)_{xy} \rangle$ of 0.2 and 0.3, and even the significantly larger values in experiments with long AP trains (Fig. 3c,g; Fig. 4c,g), these values are surprisingly large if compared to results obtained by measurements of steady-state activation parameters with long rectangular pulses. This is because also long pulses usually do not lead to true steady states⁴⁰, in particular at the low voltages near the pacemaker potential. Instead, our dynamic AP clamp at single-channel resolution seems to provide physiologically more meaningful information. For a neuron expressing mHCN2 or mHCN4V_E, our results suggest that long-lasting spike activity leads to an enhanced non-specific conductance in a use-dependent manner, enabling a stronger cAMP control on the neuronal activity. Apart from this effect of the length of AP trains, our measurements showed for all isoforms marked variability of the individual $P_o(t)_{xy}$ values (Fig. 3c,g,k; Fig. 4c,g,k; Supplementary Tables 2-7). Possible reasons for this variability are the differences of the voltage shift ΔV among the patches, which we set to a mean value (see Supplementary Table 1) and the stochastic activity of single channels per se over time. Despite this variability in the amplitude of $P_o(t)_{xy}$ among the measurements, the exclusive result of a pronounced activation and deactivation gating in mHCN1 at the slow AP frequency is remarkably consistent (Fig. 4k).

Point-by-point responses to Reviewer #3:

This paper investigates which hyperpolarization-activated cyclic nucleotide-gated (HCN) channel isoforms function as neuronal pacemaker channels during rhythmic firing. Using a dynamic clamp, the authors directly track the activity of HCN1, HCN2, and HCN4 channels during realistic pacemaker depolarizations. They show that although all three isoforms exhibit some open probability, only HCN1 gates rapidly enough to activate and deactivate within individual action potential cycles. In contrast, HCN2 and HCN4 display gating kinetics that are too slow, contributing mainly to setting the background membrane potential rather than pacing. Simulations indicate that HCN1 provides only the initial portion of the pacemaker depolarization, acting as a trigger channel before other conductances take over. The study concludes that HCN1 is the sole genuine neuronal HCN pacemaker channel, potentially redefining the functional division of labor among HCN isoforms in the brain.

The work appears to be well executed, and the highly technical paper is well written and clear. I focus my review on the simulation component, which is the area I am most familiar with.

We like to thank the reviewer for her/his appreciation of our work.

Overall, the simulations are done at a good standard and provide a useful component of the study. I would suggest that the authors move the supplementary figures about simulations to the main body of the paper and provide a short description of what they did and observed in

Results (currently it is only mentioned briefly in Discussion). With only 4 main figures in the current version, these additions should fit easily.

We followed the suggestion and moved the (new) simulations to the new Fig. 6 in the main manuscript (see below).

In terms of the simulation detail, my main concern is about using the open probability trace averaged over 12 experiments. I would recommend that the authors repeat simulations for non-averaged data from each of the 12 experiments. Fig. 3k suggests that non-averaged $P_0(t)_{s1}$ fluctuates much more and potentially reaches higher values than the averaged version. This may affect the results. It is therefore useful to use the non-averaged $P_0(t)_{s1}$ traces. It is quite likely that the main conclusions would still hold, but it would be good to check.

For the results shown in panels b, c, d of Extended Data Figure 9, it would be best to cover the whole grid of options, rather than varying one parameter at a time while holding all the others constant. With 3 parameters used here and 3 values for the first, 3 for the second, and 4 for the third, this would result in $3 \times 3 \times 4 = 36$ conditions. For each of them, it would be best to use 12 separate non-averaged $P_0(t)_{s1}$ traces, as I suggested above. That becomes $36 \times 12 = 432$ simulations. Given that all the code exists, and the simulations of a single-compartment neuron are not computationally demanding, this should be feasible without too much trouble. Having such a more complete set of simulations would provide a more convincing evidence that the main conclusions are supported (assuming the results do not change substantially).

We have revised the simulations according to the reviewer's suggestions. The results are now shown as Main Figure 6. These revised simulations confirm our previous conclusions. As varying R_{in} has no notable effect on the maximum voltage reached during the pacemaking phase (Fig. 6c), we only show all possible combinations of E_K and the scaling factor at a fixed R_{in} (Fig. 6b).

We like to thank the Reviewer for this suggestion because this new analysis based on the individual time courses better demonstrates the robustness of our observations.

The revised text parts in the Discussion read:

To further explore this observation and quantitatively estimate the effects of the HCN1 isoform on pacemaking in a neuron, we simulated membrane potential dynamics in a spherical, isopotential model SCN neuron. HCN1 conductance time courses were derived from the individual experimental P_0 measurements (c.f. Fig. 4k) to include their statistical variability (Fig. 6a). The model reveals that as a sole pacemaker conductance, HCN1 will only depolarize the neuron to at most ~ -73 mV across a wide range of simulation parameters, corresponding to only the initial $\sim 25\%$ of the total depolarization required to reach the threshold (Fig. 6b,c). To reconstruct the full depolarizing pacemaker ramp, an additional depolarizing conductance is required to take over pacemaking after this initial phase (Fig. 6d).

While previous work has established that HCN1 activates faster and at more depolarized potentials than other HCN family members, our simulations constrained by experimentally measured HCN1 gating show that, under realistic biophysical conditions, dynamic HCN gating in the physiological subthreshold range is effectively confined to HCN1, with additional HCN family conductances contributing primarily tonic background currents. In this context, our results provide a framework that may help explain why HCN1 channels are particularly suited to temporally confine subthreshold synaptic integration, a property previously implicated in the regulation of learning and memory in CA1 pyramidal neurons (References).